# Long-Term Immunogenicity and Efficacy of the Oral Rabies Virus Vaccine Strain SPBN GASGAS in Foxes

**DOI:** 10.3390/v11090790

**Published:** 2019-08-27

**Authors:** Conrad M. Freuling, Verena te Kamp, Antonia Klein, Maria Günther, Luca Zaeck, Madlin Potratz, Elisa Eggerbauer, Katharina Bobe, Christian Kaiser, Antje Kretzschmar, Steffen Ortmann, Peter Schuster, Adriaan Vos, Stefan Finke, Thomas Müller

**Affiliations:** 1Institute of Molecular Virology and Cell Biology, Friedrich-Loeffler-Institut (FLI), WHO Collaborating Centre for Rabies Surveillance and Research, OIE Reference Laboratory for Rabies, 17493 Greifswald-Insel Riems, Germany; 2IDT Biologika GmbH, 06861 Dessau-Rosslau, Germany

**Keywords:** oral rabies vaccines, rabies, genetically engineered vaccine, red foxes, efficacy, long-term immunogenicity, SPBN GASGAGS

## Abstract

To evaluate the long-term immunogenicity of the live-attenuated, oral rabies vaccine SPBN GASGAS in a full good clinical practice (GCP) compliant study, forty-six (46) healthy, seronegative red foxes (*Vulpes vulpes*) were allocated to two treatment groups: group 1 (*n* = 31) received a vaccine bait containing 1.7 ml of the vaccine of minimum potency (10^6.6^ FFU/mL) and group 2 (*n* = 15) received a placebo-bait. In total, 29 animals of group 1 and 14 animals of group 2 were challenged at 12 months post-vaccination with a fox rabies virus isolate (10^3.0^ MICLD_50_/mL). While 90% of the animals offered a vaccine bait resisted the challenge, only one animal (7%) of the controls survived. All animals that had seroconverted following vaccination survived the challenge infection at 12 months post-vaccination. Rabies specific antibodies could be detected as early as 14 days post-vaccination. Based on the kinetics of the antibody response to SPBN GASGAS as measured in ELISA and RFFIT, the animals maintained stable antibody titres during the 12-month pre-challenge observation period at a high level. The results indicate that successful vaccination using the oral route with this new rabies virus vaccine strain confers long-term duration of immunity beyond one year, meeting the same requirements as for licensure as laid down by the European Pharmacopoeia.

## 1. Introduction

Despite tremendous progress in wildlife rabies control through large-scale oral rabies vaccination (ORV) programs as exemplified by Western and Central Europe, Canada, and the US, wildlife rabies continues to be a major problem in many countries of the world with red foxes (*Vulpes vulpes*), raccoons, skunks, and raccoon dogs. These animals account for the great majority of the reported wildlife cases in the Northern hemisphere. Given its cost-effectiveness and suitability for application over large geographical areas, ORV is the method of choice to control the disease in its wildlife reservoirs [1,2,3]. A prerequisite is a vaccine that is safe and efficacious via the oral route and a bait delivery system that is suitable for the target species [4]. In Western Europe and Canada, rabies in red foxes was eliminated through large-scale ORV using live rabies virus vaccines, including ERA-BHK21, SAD Bern, SAD B19, SAG1, and SAG2 [5,6]. Although there is no doubt about the effectiveness of live-attenuated rabies vaccines of the 1st and 2nd generation under field conditions, and despite more than 750 million baits distributed between 1978 and 2018 [5,6,7], the continued use of vaccines at least of the 1st generation has been questioned [8]. To overcome limitations of those vaccines regarding safety, e.g., residual pathogenicity for rodent species [9,10,11,12,13] and observed vaccine-induced rabies in target and non-target animals [6,14,15,16,17], high genetic diversity within certain commercial vaccine strains [18,19,20], temperature stability and ineffectiveness of the oral route in rabies reservoirs such as raccoons and skunks [21,22,23]; alternative vaccines with, for instance, a higher safety profile have been developed. Recombinant vaccines constructed from heterologous virus vectors like the vaccinia virus and human adenovirus type 5 that express the RABV G [24,25,26,27,28,29] have been widely used in ORV programs to combat rabies in wildlife, in particular in foxes and raccoons in Western Europe and North America [2,30,31,32,33,34,35]. Given the availability of a reverse genetics system based on ORV SAD B19 [36], rational modification by site-directed mutagenesis and gene insertions has allowed further attenuation [37] and generated new hopes for the increased efficacy and safety of future oral rabies vaccines. Among the numerous constructs obtained from such approaches, including ERA 333 [38], SPBN GAS [39], SPBN GASGAS [40], SAD dIND [41], ORA-DPC [42,43], rLBNSE-DCBp [44], and LBNSE-CXCL13 [45], only ERA 333 and SPBN GASGAS have been successfully tested in wildlife target species [38,40].

For any vaccine, the duration of immunity (DOI) or duration of protection is an essential part of its efficacy [46]. The DOI provided by vaccines varies depending on a range of factors, particularly the vaccine itself and, therefore, information on the DOI is important to achieve optimal immune protection. In case life-long protection is not achieved, knowledge about the DOI is required when recommending the timing of appropriate vaccination intervals to cover the periods in life when the vulnerability of individual animals to the disease is highest or for defining optimal vaccination strategies at a population level. In rabies control, the latter applies to mass dog vaccination campaigns and oral vaccination of wildlife. 

While some inactivated parenteral rabies vaccines are known to produce a long DOI of at least four years in dogs and cats [47,48,49,50], for the oral rabies vaccines used today for wildlife, relatively little data is available concerning the length of time that they give protection [51,52,53]. According to the European Pharmacopoeia (Ph.Eur.), a DOI of at least six months is required for licensure, which is in line with the vaccination schedule as recommended by the EU; i.e., two oral vaccination campaigns per year (Spring, Autumn) [54]. However, to achieve cost-effective prevention of reintroduction of rabies from bordering areas, vaccination once per year as has been practiced by Turkey, Kosovo, and Finland [55,56,57] would be helpful. For this purpose, a DOI of at least 12 months needs to be confirmed using experimental data. Therefore, the objectives of this study were: (1) To quantify the DOI of captive red foxes (*Vulpes vulpes*), following ORV using baits laden with the highly attenuated rabies vaccine construct SPBN GASGAS, and (2) to test the protection conferred by the vaccine via challenge at 12 months post-vaccination according to the same requirements as outlined in the European Pharmacopoeia (Monograph 0746: Rabies vaccine (live, oral) for foxes and raccoon dogs).

## 2. Material & Methods

### 2.1. Study Animals and Housing

We purchased forty-six captive-reared foxes aged 5–6 months from a commercially registered Polish breeder. The health status of the animals was assessed and confirmed by a European Union certificate of intra-community trade. Before treatment, the foxes were de-wormed (Panacur®, MSD-Tiergesundheit, Germany) and vaccinated against canine distemper (Biocan NOVEL, Bioveta, Czech Republic). For the serology and challenge study, the foxes were housed in animal biosafety level 2 and 3 ** facilities at two investigator sites (28 animals at IDT Biologika GmbH; 18 animals at FLI). Once a day, the animals were provided with commercial standard feed (300–500 g) without any antibiotics, according to individual consumption behavior and need, except for a weekly fasting day, and water ad libitum. Occasionally, the feed was enriched with fruits, vegetables, mouse and chicken cadavers, or boiled eggs. In general, animal husbandry during the experimental studies and monitoring of the animals followed the procedures of previous efficacy studies [40] and general care was provided as required.

### 2.2. Ethics Statement

Animal housing and maintenance was in accordance with national (Tierschutz-Versuchstierverordnung-TierSchVersV) and European legislation (Directive 2010/63/EU) and followed the guidelines for the veterinary care of laboratory animals [58]. To avoid unnecessary suffering, early humane clinical endpoints were defined as described in Reference [59]. All animal studies were evaluated and approved by the responsible authorities in the federal state of Saxony Anhalt (Landesverwaltungsamt Sachsen-Anhalt, Referat Verbraucherschutz, Veterinärangelegenheiten, 06003 Halle, Germany, IDT-42502-3-761) and Mecklenburg-Western Pomerania (Landesamt für Landwirtschaft, Lebensmittelsicherheit und Fischerei Mecklenburg-Vorpommern, 18003 Rostock, Germany, FLI-7221.3-1-087/16).

### 2.3. Vaccination and Challenge

This study was a placebo-controlled, blinded study following good clinical practice (GCP) guidelines [60]. The foxes were randomly allocated to two treatment groups. Following a 24 h fasting period for both groups, group 1 (*n* = 31) received a vaccine bait containing the vaccine strain SPBN GASGAS with a titer of 10^6.6^ FFU/mL [40], while group 2 (*n* = 15) received a placebo-bait. Bait debris and vaccine spillage were collected below each cage. If the bait matrix was partially ingested but the blister was still intact after 24 h, a new bait was offered. At 53 weeks post-vaccination (p.v.), all animals were challenged intramuscularly (i.m.) with a 10^3.0^ MICLD50/dose of rabies field virus strain “fox Krefeld” (FLI ID 148), as described in Reference [40], then and monitored for another 90 days post-infection (p.i.). During the entire experimental phase, all animals were observed daily for general health. Blood samples were taken prior to vaccination (B0) and at different time points post-vaccination throughout the study, e.g., 3 (B1), 5 (B2), 9 (B3), 18 (B4), 26 (B5), 39 (B6), 53 (B7), 55 (B8), 57 (B9), and 66 (B10) weeks p.v. All surviving animals were euthanized 90 days p.i. (week 66) as described in Reference [40].

### 2.4. Diagnostic Assays and Statistical Analysis

From all survivors, as well as from the euthanized animals after challenge infection because of clinical scores, brain samples were taken and used for rabies diagnosis using the direct fluorescent antibody test (FAT), including defined positive (PC) and negative controls (NC) [61]. Reverse transcription quantitative real-time polymerase chain reaction (RT-qPCR, [62]) was used to confirm the presence of viral RNA in brain samples. A modified fluorescent focus inhibition test (RFFIT, [63]) was used to establish baseline titers for rabies virus neutralizing antibodies (VNA) prior to vaccination (B0) as well as p.v. and p.i. (B1–B10). Test performance, calculation of VNA titers, and the subsequent conversion into international units (IU) per ml following normalization against the World Health Organization 2nd International Reference Standard (National Institute of Biological Standards and Controls, Potter’s Bar, UK) was done essentially as described in Reference [63]. Animals with an antibody titer ≥0.5 IU/mL were considered seropositive. For the detection of rabies specific binding antibodies, a commercial blocking ELISA (BioPro Rabies ELISA, Czech Republic) was used as described in Reference [64]. A 40% inhibition of the test serum compared to the negative control was considered as the cut-off for seropositivity, provided that the internal validity criteria indicated in the instruction of the manufacturer were met. An animal was considered a “responder” to vaccination if they were seropositive in at least one assay at least five weeks p.v.

Differences in mean percent blocking (MPB) values and geometric mean titers (GMT) were tested for significance using unpaired *T*-tests with a significance level of α = 0.05. Survival was analyzed using Kaplan–Meyer-plots and statistical differences in survival were assessed by the Log-rank (Mantel–Cox) test as implemented in GraphPad Prism version 7.00 (GraphPad Software, La Jolla, CA, USA).

## 3. Results

### 3.1. Bait Acceptance, Vaccination, and General Health

In group 1 (vaccination group), twenty-two foxes (70%) consumed baits within 24 h, while for the remaining nine animals, bait consumption took a longer time. Four of those animals consumed baits at day 2 and 3, and for one animal bait consumption was recorded on day 4. In a further four cases, unconsumed vaccine baits were recovered on day 2 and new baits were offered, while one fox was offered a new bait on three consecutive days (days 2–4). In Group 2 (controls), ten of 15 (66.6%) foxes consumed the baits the same day, while for four and one animal bait consumption was recorded at day 2 and 3, respectively. No vaccine-induced morbidity or mortality was observed. Three animals (No. 23 from Group 2, No. 32 and 40 from Group 1) died or had to be euthanized prior to challenge infection for reasons not related to the vaccine, e.g., perforation of the small intestine or infection after self-inflicted bite wounds.

### 3.2. Immunogenicity

The mean percent blocking (MPB) value and geometric mean titer (GMT) of pre-vaccination sera for all animals and sera for all control foxes throughout the observation period p.v., as measured by ELISA and RFFIT, was ≤13.9% and ≤0.07 IU/mL, respectively. Two weeks p.v., 90.3% of the foxes (28/31) in the vaccine bait treatment group (Group 1) developed rabies virus-specific binding antibodies with an MPB value of 74%. Eighty-one percent (25/31) of animals showed VNA titers of ≥0.5 IU/mL at the same time point. There was a slight increase in both the MPB and GMT values from week three p.v. until the date of challenge (Figure 1, Appendix A). For all sampling dates (except B0), the MPB and GMT values for Group 1 foxes throughout the pre-challenge phase were 84.4% and 1.66 IU/mL, respectively. In total, 89.7% of the foxes (26/29) in the vaccination group (Group 1) were classified as responders. At the time point of challenge, in five animals (No. 3, 14, 17, 24, and 44), the VNA titers had dropped below the threshold of 0.5 IU/mL, whereas only one animal (No 17, Appendix A) was also negative in ELISA. Additionally, the three vaccinated animals (No. 15, 34, and 41) that succumbed to challenge infection never demonstrated any VNAs nor binding antibodies throughout the study (Appendix A).

Post challenge, vaccinated animals responded with a strong VNA titer increase to a GMT of 15.9 IU/mL within seven days, in contrast to a less pronounced increase in the MPB to 98.0%. VNA titers then declined slightly by 90 days p.i. (week 66), the end of the experiment, while MPB values remained almost constant. Additionally, all previously seronegative animals demonstrated rabies specific antibodies after challenge, at the time point of rabies-related euthanasia (Figure 1).

### 3.3. Resistance to Rabies Challenge

Of the 29 vaccinated foxes (Group 1) challenged 53 weeks p.v., 26 (89.6%) survived the infection (Figure 2). If we only considered foxes classified as responders to vaccination by seroconversion at B2, the efficacy of the vaccination was 100%. On the other hand, 13 of the 14 (93.3%) control foxes that were challenged had to be humanely euthanized after showing clinical signs, with rabies being confirmed by detection of RABV antigen (FAT) or viral RNA (RT-qPCR). The incubation period in this experimental setting (time, until the animal was euthanized, usually the same day on which the first clinical signs were observed) ranged between 12 and 17 days p.i., with a median survival of 15 days p.i.

## 4. Discussion

In a previous study, the efficacy of the genetically engineered SPBN GASGAS rabies virus vaccine strain was demonstrated in raccoon dogs and foxes six months after vaccination [40]. The study was the first to examine the long-term immune response of red foxes to this vaccine after bait application. While a 6-months DOI is the minimum requirement for licensure according to the guidelines of the Ph.Eur. [65], a longer DOI of 12 months after bait delivery is requested in the US (Code of Federal Regulations, Title 9, Animals and Animal Products (9CFR) [66]. The results of this study supported the claim that the rabies strain SPBN GASGAS confers protection for at least one year after vaccination via bait. 

Depending on where the vaccines were licensed, a DOI of at least 6 to 12 months after vaccine delivery was shown for attenuated rabies virus vaccines, e.g., ERA, SAD B19, SAD P5/88, SAG2 [7,53,67,68,69], vector-based recombinant vaccines including VRG [70,71,72] and human adenovirus type 5 rabies recombinant vaccine (ONRAB) [4,73,74], as well as reverse genetic constructs such as ERA333 and SPBN GASGAS [38,40]. Demonstration of DOI for the oral rabies vaccines to be used in dogs under laboratory and field conditions is also recommended [75]. For example, for the oral vaccine constructs SAG2 [76] and VRC-RZ2 [77] there is a DOI of 6 months, while for the vaccine candidate CAV-2-E3Δ-RGP [78] a DOI of 2 years after vaccine bait administration was demonstrated in dogs.

The intensive monitoring and sampling of the animals allowed a close investigation of VNA and binding antibody kinetics as measured by RFFIT and ELISA. The results proved the induction of a specific immune response at three weeks p.v., which remained at a very high level up to 12 months p.v. The strong increase in rabies antibody levels immediately after challenge was the result of a booster effect to the challenge virus (Figure 1), as also observed in other efficacy studies [67,74,79,80]. In other studies, no indication of a booster effect after administration of the challenge virus could be found [68,69]. However, in the latter studies, blood samples were only tested 90 d.p.i., thereby likely missing the temporal increase in titers.

The less pronounced increase in MPB values p.i. compared to the VNA titers results from the nature of the ELISA test used, because the saturation of the blocking capacity of individual sera had already been reached in the pre-challenge phase (Figure 1). Similar antibody kinetics were achieved in a previous study, where foxes were immunized with the same dose of the vaccine, but for reasons unknown, had mounted a much lower immune response [40].

In this study, there was a high agreement of seropositivity and survival (Figure 2). All “responders” (seropositive in at least one test 5 weeks p.v.) to vaccination survived, while none of the animals, which had to be euthanized had developed any measurable response to the vaccination. These animals had been declared as vaccinated, but in two cases, the bait and blister were not discovered. The lack of a measurable immune response suggests that contact with the vaccine may not have taken place. On the opposite, only one control animal (No. 17) without rabies specific antibodies survived, whereas one vaccinated animal (No. 44) with a GMT below the threshold of 0.5 IU/mL during the entire pre-challenge phase also survived. However, this animal was positive in the ELISA (Appendix A), confirming a previous statement that the BioPro ELISA is a better predictor for survival than the RFFIT [63]. Together, these data demonstrate that SPBN GASGAS vaccinated animals, which seroconverted five w.p.v. as confirmed by ELISA, survive a challenge infection even one year after vaccination. It is likely that general seroconversion and not the titer at challenge infection determines the outcome of challenge infection, as discussed earlier [51,53,63]. In a broader perspective, since the positive serological response to vaccination, as measured by ELISA, was 100% correlated with survival, this marker appears to be a very robust surrogate for protection. Implementing this approach into licensing guidelines and requirements would render the use of naïve controls in challenge infections dispensable, thereby reducing unnecessary animal suffering.

In the present study, three foxes that were offered a bait did not show a detectable immune response. It is a well-known fact that vaccine administration by baits is less efficient than transdermal administration by injection. Often, tetracycline is incorporated into the bait matrix as marker to determine bait uptake by the target species during oral vaccination campaigns. During post-campaign monitoring, most of the time, a discrepancy of 10–20% is found between the bait uptake rate as determined by the presence of the bait marker and vaccination coverage as determined by the presence of rabies antibodies [81,82,83]. This difference is predominantly the result of bait handling; sometimes the animals swallowed the sachet without perforation so that the vaccine is not released in the oral cavity or the animals separate the sachet from the bait matrix. Another possibility is that most of the vaccine is spilled on the ground during bait handling. As a result, the vaccine is not taken up in the oral cavity and presented to the immune system of the animals, and consequently no immune response is developed. For two animals, no blister was recovered after bait consumption; hence, it cannot be ascertained that the vaccine was released in the oral cavity. Extended spillage was observed with the third fox. Many different challenge studies involving an oral rabies vaccine bait have encountered similar problems [4,73,74,84]. The fact that bait handling by the animals cannot be controlled during these studies is problematic. In a direct comparison, it was shown that 100% and 81% of skunks receiving the same oral rabies vaccine and dose by direct oral instillation or by offering a vaccine bait, respectively, survived a challenge infection [73]. Sometimes, it is possible to observe the animals and differentiate, for example, between animals with no or partial spillage of the vaccine virus, which can be incorporated into the analysis and interpretation of the results obtained [84].

Efficacy is defined as the ability of an intervention to produce the desired beneficial effect, and vaccine efficacy is the percentage reduction of disease in a vaccinated group compared to an unvaccinated group using the most favourable conditions [85]. As pointed out, efficacy studies of oral rabies vaccines by offering the animals’ vaccine baits do not represent ideal conditions, since bait handling by the animals is not under direct control. Therefore, a distinction should be made between vaccine and vaccine bait efficacy, whereby the former can be tested by direct oral application (optimal conditions) and the latter by offering the animals a vaccine bait (sub-optimal conditions). Presently, the efficacy requirements for oral rabies vaccine baits, in terms of the minimum proportion of vaccinated and control animals that survive and succumb to challenge infection, respectively, are identical to parenteral rabies vaccines for pets like dogs and cats. Considering that the safety requirements for oral rabies vaccines targeted at wildlife are more stringent than for rabies vaccines targeted at pets and livestock, owing to the unique distribution system, the question of whether the efficacy requirements of oral rabies vaccines for wildlife should be adjusted is justified. Vaccination campaigns provide both direct and indirect protection against infectious diseases. Direct protection occurs by lowering or eliminating (sterile immunity) the probability that vaccinated animals become infected, whereas indirect protection represents the reduction of the transmission rate for both vaccinated and unvaccinated individuals within the population [86].

Vaccination of wildlife against rabies should prevent the spread of infection through herd immunity; thus, the protection of every individual animal against infection is not necessary. Protection should be sufficient to reduce onward transmission to other animals below a threshold, resulting in the interruption of the transmission cycle. For rabies, both in wildlife and dogs, vaccination coverage of 70% is set as a target to successfully and efficiently interrupt the transmission [87,88]. Therefore, the direct effect of vaccination in addressing the individual animal is less relevant than the overall effectiveness of the oral rabies vaccination campaigns considering indirect protection of unvaccinated individuals in a partially vaccinated population. Hence, it is suggested to adjust the minimum efficacy requirements for oral rabies vaccine baits, taking the efficiency loss during bait handling into consideration.

## 5. Conclusions

This study demonstrates that the rabies virus strain SPBN GASGAS induces a long-lasting humoral immune response and confers protection after a highly virulent RABV challenge for at least 12 months after application in foxes following the same requirements as for licensure (European Pharmacopoeia monograph No.0746/2014). 

## Figures and Tables

**Figure 1 viruses-11-00790-f001:**
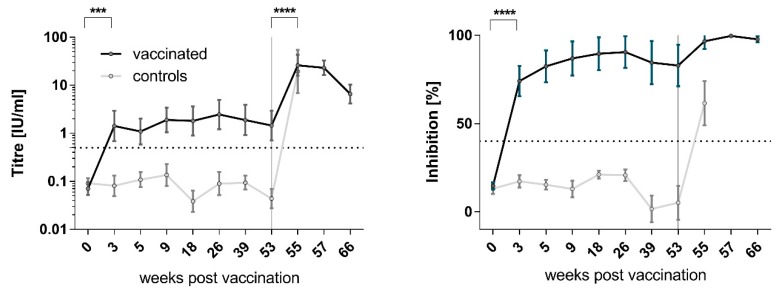
Development of immune response p.v. and p.i. as measured by RFFIT (left) and ELISA (right) for vaccinated (black line) and control foxes (grey line). For better visualization, the ELISA values (percent inhibition) and VNA titers (IU/mL) per sampling are displayed as mean and GMT, respectively, including the standard deviation. The dotted lines represent the cut-off and threshold of positivity in RFFIT (0.5 IU/mL) and ELISA (40%). There was a significant increase in VNA titers of vaccinated animals between B0 and B1 (unpaired *t*-test, *α* = 0.05, *p* < 0.0002) and post-challenge between B7 and B8 (*p* < 0.0001), whereas for ELISA, only the difference in the mean inhibition between B0 and B1 was significant (*p* < 0.0001).

**Figure 2 viruses-11-00790-f002:**
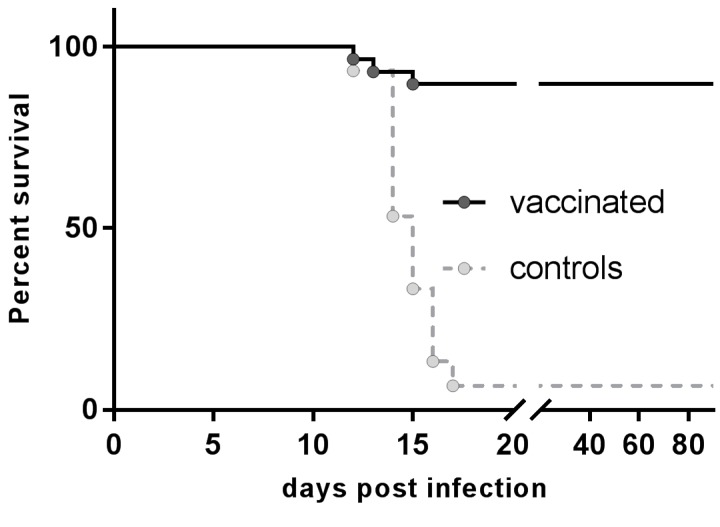
Survival curves of vaccinated (solid line) and control (dashed line) foxes. Foxes were challenged 53 weeks p.v. (day 0 p.i.) and observed for 90 days (66 weeks p.v.). The median incubation period for control foxes was 15 days (range: 12–17 days). Ninety percent of foxes offered SPBN GASGAS baits survived the RABV challenge infection, whereas only one of 15 control animals (7%) resisted the challenge infection, representing a significant difference in survival between both groups (Log-rank (Mantel–Cox) test, *p* < 0.0001).

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
