# Peer review of "Long-Term Immunogenicity and Efficacy of the Oral Rabies Virus Vaccine Strain SPBN GASGAS in Foxes"

_viruses, 2019, doi:10.3390/v11090790_

Round 1

Reviewer 1 Report

Line 117 the term "essentially" is used to describe the method of euthanasia.  It is unclear if the method used is as described in citation 40 or if the method used was "essentially" as in citation 40.  That is, was there a modification.  If there was a modification the change should be noted.  The term "essentially" can have duplicate meanings, leading to confusion.  Also see line 128, similar use of the word.

line 108: ad 's after month to read ...12 months after application...

Author Response

Line 117 the term "essentially" is used to describe the method of euthanasia.  It is unclear if the method used is as described in citation 40 or if the method used was "essentially" as in citation 40.  That is, was there a modification.  If there was a modification the change should be noted.  The term "essentially" can have duplicate meanings, leading to confusion.  Also see line 128, similar use of the word.

A: The reviewer is correct and the word essential was removed.

 line 108: ad 's after month to read ...12 months after application...

A: Changed as suggested.

Reviewer 2 Report

The co-authors present data regarding the efficacy of oral rabies vaccine SPBN GASGAS in foxes one year post-vaccination.

I have no major concerns with the manuscript.

Minor comments/edits

line 15: immunity of the live-attenuated, oral rabies vaccine

line 17: itilicize Vulpes vulpes

line 28: requirements as for

methods section 2.3: it's not clear for animals that did not accept the vaccine bait on day 1 was fasting continued?

line 153-154: this statement is somewhat confusing maybe reword or add a reference to B0 to help clarify. Part of the confussion could be related to the first mention of the 57-week observation/acclimation period. Sepearating out this statement might help the reader.

line 164-166: this statement appears to contridict the data in Table 1. Table 1 shows 100% seroconversion by the ELISA but if three animals never demonstrated binding antibodies then I think seroconversion should be 90%. Please clarify this statement or correct Table 1.

line 170-171: I assume this statement applies to both control and vaccinated animals. Also did the one control animal that survived seroconvert post-infection?

Figure1: error bars for RFFIT (vaccinated, up) and ELISA (controls, down) appear to have been lost in formating.

pg. 7 line 33: appears to be an extra space before "in dogs."

pg. 8 line 38, 53: Orciari, et al. Vaccine (2001) showed that dogs vaccinated with SAG-2 that did not have a detectable antibody titer before challenge survived if they demonstrated a rapid anamnestic response to challenge; thus the pre-challenge titer is less important than the post-challenge response.

pg. 9 line 99: It may be helpful to cite the 70% vaccination coverage in dogs that is predicted to interupt transmission of canine rabies.

pg. 9 line 110: What is meant by "ELISA was almost 100% correlated"? Either it was or it was not 100%. Please enter the actual % correlation. Also lines 109-113 should be moved to discussion since this correlation was not previously discussed or presented in results.

Author Response

line 15: immunity of the live-attenuated, oral rabies vaccine

Changed

line 17: itilicize Vulpes vulpes

Changed

line 28: requirements as for

Changed

methods section 2.3: it's not clear for animals that did not accept the vaccine bait on day 1 was fasting continued?

A: No, animals received food and water. The text was slightly modified accordingly

line 153-154: this statement is somewhat confusing maybe reword or add a reference to B0 to help clarify. Part of the confussion could be related to the first mention of the 57-week observation/acclimation period. Sepearating out this statement might help the reader.

A: There was a mistake in the 57-week. Challenge took place at week 53 post vaccination.

line 164-166: this statement appears to contridict the data in Table 1. Table 1 shows 100% seroconversion by the ELISA but if three animals never demonstrated binding antibodies then I think seroconversion should be 90%. Please clarify this statement or correct Table 1.

A:There was a formatting error in Table 1 which caused the discrepancies. Against the background of i) that all information is provided in the text, or in figures, and ii) that all raw data is available in the supplementary Table, we opted to remove Table 1 altogether. In the supplement, we included the rabies diagnosis (i.e. FAT result) and the incubation period.

line 170-171: I assume this statement applies to both control and vaccinated animals. Also did the one control animal that survived seroconvert post-infection?

A: Yes, all animals seroconverted post challenge.

Figure1: error bars for RFFIT (vaccinated, up) and ELISA (controls, down) appear to have been lost in formating.

Changed

7 line 33: appears to be an extra space before "in dogs."

Changed

8 line 38, 53: Orciari, et al. Vaccine (2001) showed that dogs vaccinated with SAG-2 that did not have a detectable antibody titer before challenge survived if they demonstrated a rapid anamnestic response to challenge; thus the pre-challenge titer is less important than the post-challenge response.

A: Well, this may be another issue in animals that did not show a measurable immune response but were protected. Here, we discuss the outcome of animals which did show an anamnestic response to vaccination and were protected. While we can measure the pre-challenge titer, and, as we believe, as early as 3 weeks p.v. with a positive result indicating protection against challenge, the post-challenge response is outside of the diagnostic window.

9 line 99: It may be helpful to cite the 70% vaccination coverage in dogs that is predicted to interupt transmission of canine rabies.

A: The text was amended and reference was given to the 70% threshold.

9 line 110: What is meant by "ELISA was almost 100% correlated"? Either it was or it was not 100%. Please enter the actual % correlation. Also lines 109-113 should be moved to discussion since this correlation was not previously discussed or presented in results.

A: We changed this part of the conclusions and moved part into the discussion as suggested.